# Response to Mechanical Properties and Physiological Challenges of Fascia: Diagnosis and Rehabilitative Therapeutic Intervention for Myofascial System Disorders

**DOI:** 10.3390/bioengineering10040474

**Published:** 2023-04-14

**Authors:** Yuya Kodama, Shin Masuda, Toshinori Ohmori, Akihiro Kanamaru, Masato Tanaka, Tomoyoshi Sakaguchi, Masami Nakagawa

**Affiliations:** 1Department of Orthopaedic Surgery, Okayama Rosai Hospital, 1-10-25 Midorimachi, Minamiku, Okayama 702-8055, Japan; me18066@s.okayama-u.ac.jp (S.M.); sooseizi0402@yahoo.co.jp (T.O.); akihiro198326@gmail.com (A.K.); tanaka0896@gmail.com (M.T.); 2Department of Central Rehabilitation, Okayama Rosai Hospital, 1-10-25 Midorimachi, Minamiku, Okayama 702-8055, Japan; tomoyoshi0127@gmail.com (T.S.);

**Keywords:** fasciae, innervation, injury, hyaluronic acid, aging, sex hormone, rehabilitation, imaging, myofascial release

## Abstract

Damage to the fascia can cause significant performance deficits in high-performance sports and recreational exercise and may contribute to the development of musculoskeletal disorders and persistent potential pain. The fascia is widely distributed from head to toe, encompassing muscles, bones, blood vessels, nerves, and internal organs and comprising various layers of different depths, indicating the complexity of its pathogenesis. It is a connective tissue composed of irregularly arranged collagen fibers, distinctly different from the regularly arranged collagen fibers found in tendons, ligaments, or periosteum, and mechanical changes in the fascia (stiffness or tension) can produce changes in its connective tissue that can cause pain. While these mechanical changes induce inflammation associated with mechanical loading, they are also affected by biochemical influences such as aging, sex hormones, and obesity. Therefore, this paper will review the current state of knowledge on the molecular level response to the mechanical properties of the fascia and its response to other physiological challenges, including mechanical changes, innervation, injury, and aging; imaging techniques available to study the fascial system; and therapeutic interventions targeting fascial tissue in sports medicine. This article aims to summarize contemporary views.

## 1. Introduction

Injuries to various fascial tissues can significantly reduce performance in sports [1] and may contribute to the development and persistence of symptoms of musculoskeletal disorders, including low back pain [2]. Although damage to the fascial tissue has been shown to be influenced primarily by exercise, aging, sex hormones, obesity, and inflammation [3,4,5,6,7,8,9,10,11,12,13,14], there is limited information about the elastic fiber composition of fascial tissue, extracellular matrix properties, vascularity, extent of innervation, and their role in disease and treatment [15]. Although these properties are fundamental for optimal sports performance, the fascia is not considered in the physical examination and rehabilitation of sports injuries, omitting anatomical tissues that could play an important role in sports injury rehabilitation. For example, acute pain associated with the groin is common in athletes injured by overuse. Acute strains have been shown to occur at the musculotendinous junction, particularly in the adductor longus, rectus femoris, and iliopsoas muscles [16]. X-rays and MRIs are performed to rule out the diagnosis of serious diseases, such as fractures [17,18,19,20], and if serious diseases are ruled out, pain provocation tests are performed using palpation, stretching, and resistance testing [21,22]. However, these evaluation methods focus on the joint range of motion and muscle strength and do not consider the assessment of disorders of the fascia [23,24].

An accurate understanding of the mechanical behavior of the fascia plays an important role in investigating pathological phenomena and comprehensive analysis of its functionality. Furthermore, the fascia and muscle in sports injuries can be identified using conventional imaging modalities such as magnetic resonance imaging (MRI) and ultrasound imaging, which are valuable guides for appropriately rehabilitating sports injuries. Furthermore, physical examination of these tissues at different sites, depending on the pathology, may influence the rehabilitation outcome. This article will review the current state of knowledge on the molecular to mechanical properties of fascia and its response to other physiological challenges, including mechanical changes, innervation, injury, and aging; ultrasound imaging and MRI available to study the fascial system; and therapeutic interventions targeting fascial tissue in sports medicine. The purpose of this article is to summarize contemporary views on the topics highlighted in the following sections.

## 2. Role of Fascial Tissue and Pathological Reactions

Fascia is widely distributed from head to toe; it encases and permeates muscles, bones, blood vessels, nerves, and internal organs, constitutes various layers of different depths [25], and is a connective tissue composed of irregularly arranged collagen fibers, clearly different from the regularly arranged collagen fibers found in tendons, ligaments, or periosteal sheets [26]. Additionally, it supports important functions of the human body, such as posture, movement, and homeostasis [25,26,27,28], and also contains various sensory receptors for proprioception, nociception, and even hormones [28].

Structurally, fascial tissues are composed of various cell types (fibroblasts, myofibroblasts, myofascial cells, and telocytes), as well as fibrous (type I and type III collagen fibers, elastin, fibrillin), aqueous (a complex mixture of water and glycosaminoglycans) components, and neural elements (free nerve endings and mechanoreceptors) [29,30]. The fact that the fascia can transmit tension far is the basis of the “biotensegrity” framework [31,32]. Biotensegrity is the application of the principle of tensegrity to the understanding of human movement, where tensegrity is an architectural principle according to which a structure (or tensegrity system) is stabilized by continuous tension with discontinuous compression and functions as a single structure [32]. As the tension in the fascia increases, the connective tissue can disperse the force around it and propagate it along the fascial system [31,32,33,34]. Forces passively imposed on the muscle by stretching are distributed throughout the tissue via the intramuscular connective tissue [33,34]. Fascia transmits tension, influences other muscles, plays a role in the proper coordination of body movements, and can reflect the direction of force vectors. Fascia can actively contract, and changes in tension are caused by contractile cells [35]. Myofibroblasts are present in developing and normal adult tissues and are responsible for altering tissue tension [35]. Normal fibroblasts are highly sensitive to physical stimuli.

The transition from fibroblast to myofibroblast is influenced by mechanical stress. Upon mechanical tension, fibroblasts differentiate into proto-myofibroblasts, which contain actin stress fibers in their cytoplasm that terminate in a fiber bundle adhesion complex [25,36,37]. The adhesion complex bridges the internal cytoskeleton and integrins of myofibroblasts with extracellular matrix (ECM) fibronectin fibers. Thus, this allowed contractile forces to be generated in the nearby ECM when traction is applied; moreover, forces within the ECM are maintained over time and are further enhanced by remodeling and collagen deposition [37]. In addition, chronic strain, such as sitting or overuse of muscles [38,39,40], infection and inflammation [40], and immobilization of the limb by trauma, fracture, or casting [28,29,30], can produce further contraction of myofibroblast smooth muscle actin fibers and contribute to joint contractures. These environments make it difficult to maintain a relaxed state, resulting in decreased mechanical tension, and consequently, myofibroblasts either dedifferentiate or undergo apoptosis [37]. The tipping point between exercise and rest is unknown; however, multiple repetitions of the contraction cycle may result in graded and irreversible tissue contraction [37].

## 3. Factors Influencing the Pathological Development of Fascia Tissue (Nerve, Disorder, Aging, Sex Hormone)

Since blood vessels and nerves are scattered throughout the fascia, invasion of these structures via fascial changes is common [37,38]. Tissue changes translate into changes in the mobility of the nerve, resulting in a decrease in the independence of the nerve from its surroundings [39,40]. Changes in the fibrous tissue around the nerve can cause entrapment lesions [39,40], and patients may notice numbness, dysesthesia, and pain [41]. Existing histological studies indicate that the fascia is the largest sensory organ because of its large surface area [42,43]. However, it has been shown that the type and density of innervation, including receptors, varies depending on the fascia present in the different parts of the body, indicating that this tissue is more complex than imagined [44,45,46,47,48,49,50,51]. The role of the fascial tissue as a sensory receptor and the environmental changes associated with structural changes in fascial tissues may increase the stimulation of free nerve endings in the fascia, potentially increasing inflammation and pain. Reportedly, free nerve endings have nociceptor characteristics, and the density and length of nociceptors increase in inflamed thoracolumbar fascia [49,50]. Moreover, repetitive identical postures, sports, and repetitive motion may produce movement patterns that increase tissue thickness and limit slippage between fascial layers. In addition, structural changes in the fascia that occur after fascial adhesions due to trauma, overuse, or surgery can alter the activation of nerve receptors embedded in the fascia [52,53,54]. However, few effective treatments are available due to the complexity of the mechanism [55,56,57].

Mechanical stress induces the release and activation of molecules stored in the ECM, triggering the cleavage products of collagen XVIII and other basement membrane components; because the ECM is the primary site of the inflammatory response that occurs in tissues. Moreover, the ECM interacts with immune cells, especially under dynamic conditions such as growth and regeneration, and the fascial tissue requires significant changes in the local ECM microenvironment to allow cellular adaptation and remodeling of the ECM [58,59,60].

Following acute injury due to overload or anoxia in fascial tissues, the immune response aims to phagocytose injured cells. The acute inflammatory response is usually brief and reversible. It involves the release of inflammatory cytokines from injured cells and macrophages, other substances (such as bradykinin, substance P, and proteases), and molecules that sensitize nociceptive nerves [61] and promotes immune cell infiltration. However, prolonged or repeated loading results in persistent inflammation [62,63,64], and the long-term presence of macrophages and cytotoxic cytokines in and around the tissue ultimately leads to progressive tissue damage. Cytotoxic cytokines (e.g., interleukin-1β, tumor necrosis factor (TNF), transforming growth factor-β (TGFβ-1)) promote fibrosis through fibroblast overgrowth and collagen matrix deposition [65]. Notably, cytokine overproduction can also lead to nociceptive afferent nerve maintenance sensitization and increase the production and release of substance P (nociceptive neuropeptide). Prolonged or repeated loading results in persistent inflammation and the prolonged presence of macrophages and cytotoxic cytokines in and around the tissue [63,64]. Eventually, tissue damage progresses, and overproduction of cytokines is triggered. This overproduction of cytokines maintains sensitization of nociceptive afferents and increases the production and release of substance P [65]. Substance P stimulates TGFβ-1 production by fibroblasts. Furthermore, the substances P and TGFβ-1 have been shown to induce fibrogenic processes independently [65]. Therefore, it is suggested that neurogenic processes (substance P) and loading/repair processes (TGFβ-1) may contribute to increased collagen in fascial tissue. Fibrosis around fascial tissue affects secondary dynamic biomechanical properties, anchoring structures to one another or inducing chronic compression [60]. In addition, inflammatory cytokines leak into the bloodstream, causing extensive secondary tissue damage and impaired function of central nociceptors [62,66,67]. Therefore, it is suggested that the pathogenesis of myofascial tissue injury is related to the maintenance of musculoskeletal function during daily life and exercise in the elderly and the prevention of overuse injuries in athletes. The following methods have been reported to decrease inflammatory cytokines. Early treatment with anti-inflammatory agents can prevent or reduce pain induced by TNF signaling and decrease downstream collagen production [68]. Stretching of the fascial tissue can promote the resolution of inflammation both in vivo and in vitro, and manual therapy can prevent overuse-induced fibrosis in some fascial tissues [69,70]. Nevertheless, there is limited information on exercise-induced changes in myofascial tissue and its molecular response to aging. Hence, further research is needed.

Physiological aging is a highly individualized process characterized by the progressive degeneration of tissues and organ systems. For example, a sedentary lifestyle and repetitive overuse of muscles with a limited range of motion can lead to myofascial pain syndrome (MPS), producing pain in multiple areas of the musculoskeletal system resulting from myofascial changes/fibrosis. In fact, physically active workers are less likely to develop symptoms of MPS compared to sedentary workers [67]. Functionally, these pathological changes alter the mechanical properties of myofascial tissue and skeletal muscles, causing pain and age-related decreases in muscle strength and range of motion that cannot be explained by muscle mass loss alone [5]. Physical inactivity is more likely to cause pain due to collagen changes in fascia, decreased fascial sliding due to hyaluronic acid (HA) aggregation, increased contractility of myofibroblasts, and increased production of inflammatory cytokines [46,63,64,65]. Furthermore, aging is associated with fluctuations in fascia thickness. Indeed, age-related modifications are specific to different body regions. Fascia thickness in the lower extremities decreases with age (−12.3–25.8%), while that in the lower back increases (+40.0–76.7%) [71]. These connective tissue changes have been suggested to decrease joint flexibility [72]. Moreover, structural and molecular changes in fascia with aging affect force transmission in the locomotor system [73]. Fascial tissue becomes denser, and fibrosis develops with age, reducing muscle force transmission and joint range of motion [74,75]; consequently, body pain due to MPS is often attributed to age-related loss of mobility and can be viewed as a natural consequence of a sedentary lifestyle.

Today, the percentage of women participating in physical activity and sports has never been higher [76,77]. There is also a growing awareness of the potential effects of cyclical menstrual hormones (estrogen and progesterone) on exercise performance [78] and metabolic demand [79]. Estrogen is a major regulator of the female body composition; however, it is also involved in muscle damage and recovery [80,81]. Reportedly, markers of muscle damage (creatine kinase) and inflammation (interleukin 6) are significantly greater in the follicular phase than in the mid-luteal phase [82]. Thus, the menstrual cycle phase may be involved in inflammation and recovery. Therefore, contraceptive methods that can alter hormonal fluctuations, such as oral contraceptives, are sometimes used by female athletes [83,84,85]. Oral contraceptives generally contain estrogen and progestin [86] and should be cautiously used. Reportedly, females taking oral contraceptives have a lower anterior cruciate ligament (ACL) elasticity than those not taking oral contraceptives [87], and muscle-tendon stiffness in the lower extremities of young females is lower during the ovulatory phase [88,89]. However, oral contraceptives may adversely affect inflammatory markers, as evidenced by the fact that Olympic-level elite female athletes using oral contraceptives had higher levels of C-reactive protein, a marker of inflammation, than amenorrheic athletes, suggesting increased muscle damage and poor recovery potential [90,91]. In fascial tissues, acute and chronic loading stimulates collagen remodeling [6]. Moreover, the increase in collagen synthesis with exercise is lower in females than in males, and sex differences in injury frequency and estrogen receptor expression in human fascial tissue suggest that estrogen may play an important regulatory role in ECM remodeling [6,7,8]. Estrogen replacement in older postmenopausal females inhibits collagen synthesis during exercise; in contrast, it has a stimulatory effect on collagen synthesis at rest [9]. Furthermore, estrogen and estrogen receptor beta (Erβ) inhibit fibrosis by decreasing TGFβ expression, connective tissue growth factor production and function, matrix metalloproteinases 2 and 9 expressions and activity, fibroblast conversion to myofibroblasts, and collagen I and III production [89,90]. Thus, long-term estrogen deficiency is known to be associated with increased fibrosis. The presence of sex hormone receptors in fascial tissues may help explain sex differences in the prevalence of myofascial pain [88].

Changes in the physical and chemical properties of hyaluronic acid (HA) are associated with changes in the viscoelasticity, mechanical plasticity, and nonlinear elasticity of the extracellular matrix, as previously described [92,93], which may contribute to fascial disease. HA occurs between the deep fascia and muscle, facilitating sliding between these two structures and within the loose connective tissue of the fascia, ensuring smooth sliding of adjacent fibrous fascial layers [94]. Although exercise promotes HA production and recycling, the biomechanical properties of free connective tissue may change in response to the amount of lactic acid accumulated after intense exercise. Furthermore, a decrease in pH due to lactate accumulation increases the viscosity of HA, resulting in instant stiffness [95]. In contrast, immobility increases the concentration of HA without effective HA recycling, which may increase viscosity and decrease lubrication and sliding of connective tissue and muscle layers [96]. In addition, the thickening of the fascia caused by aging [66] increases the distance between surfaces and leads to an increase in HA viscosity [97]. The increase in HA viscosity within connective tissue may inhibit the sliding of fascial collagen fibers between layers [98]. The patient may perceive this increase in overall fascial thickness as increased stiffness and pain. An important component of pain treatment is reversing these changes in HA; when HA becomes sticky rather than lubricated, densification of the fascia occurs, and the distribution of force lines within the fascia is distorted [67]. Additionally, small repetitive movements, immobility, or overuse syndromes of movement that result in negative modifications of loose connective tissue can distort loose connective tissue between fascial layers and densification. From the properties of HA within the ECM, this change, reportedly, is reversible by modification of temperature, pH, and mechanical loading (such as massage) [74,98].

## 4. Imaging Diagnosis

Ultrasound and MRI are the commonly used imaging modalities for fascial injuries. Ultrasonography of fascia allows the deep fascia to be observed and provides a more convenient assessment of subcutaneous and perimuscular connective tissue thickness compared with other imaging modalities. Moreover, ultrasonography can also provide dynamic images. This allows us to identify slippage and movement between the muscle and the adjacent fascial layer and on the fascial side for the movement we wish to evaluate. Another feature is that the thickness of the fascia varies with age [99,100,101]. The thickness of the fascia and its relationship with the underlying muscles distinguish aponeurotic fasciae from epimysial fasciae, which are covered by a fibrous sheath and maintain their position as a muscle group or broad muscle attachment; in contrast, epimysial fasciae are covered by a fibrous sheath and maintain their position as a muscle group or broad muscle attachment. The epimysial fasciae characterize each muscle and determine its shape and volume. By observing each, the relationship between muscle, fascia, bone, and surrounding soft tissues can be identified [102,103]. Ultrasonography is also useful for muscle atrophy and fascial tears. Muscle atrophy can be observed by measuring the volume under ultrasound. In contrast, fascial tears can be identified by observing the damage to the continuity of the fascia. Observing the fascia while moving it and assessing the proximal and distal areas of the tear can also determine if the fascia is completely or incompletely torn [104,105]. MRI can assess fascial thickening and signal changes, adjacent soft tissue, and bone marrow edema; T1, T2; moreover, fat suppression images on MRI can identify various injuries [106]. Furthermore, MRI can identify fascial thickening, perifascial fluid retention, contiguous tears, adjacent soft tissue edema, and bone marrow edema at fascial bone marrow edema in the adherent area. Fascial thickening and complete tears are easily identified on MRI. For example, fibrous tissues may demonstrate a low signal in the T1- and T2-weighted images and a high signal in fat suppression in some cases. Acute myofascial tears show low signal areas of tear, high signal on fat suppression, and intermediate signal changes on T1. High signal areas may indicate fluid retention in the soft tissues surrounding the injury. The T1- and T2-weighted images showing heterogeneous signal intensity may indicate a neoplastic lesion. In the case of a foreign body reaction, the incidence of a low signal on T1-weighted images and a high signal on T2-weighted images is high. In addition, an infection may be present with a high signal on fat suppression and a low signal on the T1-weighted images, indicating contrasting enhancements. Thus, the degree of inflammation, localization, and spread of infection can be evaluated [106,107]. In addition, ultrasonography and MRI are easy to identify the fascia, and the contrast between the fascia and its surroundings is clear; therefore, there is a high degree of inter-specialty reliability in evaluating the fascia.

## 5. Myofascial Release (MFR) for Muscle and Fascia Dysfunction

Myofascia degenerates due to various causes. The main causes are circulatory failures due to trauma or reduced physical activity, disuse syndrome, overuse syndrome due to repetitive motion, and chronic poor posture. This causes the densification of the fascia because of the twisted collagen fibers, which harden the substrate because of dehydration. In addition, sustained muscle contraction, such as in overuse syndrome, causes hyaluronic acid aggregation, which is a factor that reduces the sliding properties of the fascia [58,69,70]. Furthermore, it has also been hypothesized that because of the continuity of fascia, dysfunction of fascia in one part of the body can cause stress in other body parts [108]. Thus, gliding between the fascia and its deeper tissues, such as muscle tissue, is inhibited, reducing the ability to maintain an antigravity posture and efficient athletic performance. Therefore, it causes poor performance in sports and interference with daily activities. Training is necessary to improve performance, especially in athletes; however, inadequate rest periods can cause high-frequency, high-intensity training that leads to continued pointless training [109]. Maladaptive training before tissue recovery and rebuilding can lead to the accumulation of microdamage in affected tissues, resulting in overuse injuries, thereby compromising the athletes’ competitive performance due to pain and dysfunction [110]. As evidenced by the fact that 39% of athletes experience unexplained musculoskeletal pain weekly [111], athletes may continue to train despite the risk of disability. Since it is difficult to determine training at the appropriate load and rest periods, we believe that MFR, as discussed below, can reduce the risk of overuse injury. MFR aims to improve the mobility and length of myofascial tissue, thereby reducing pain and allowing it to function normally [112]. As a treatment, the injection of local anesthetics into the interfascial space was expected to be a new anesthetic technique [113]. Nevertheless, it has since been clinically applied and used as a hydrorelease. Reportedly, the hydrorelease has shown therapeutic efficacy, including improvement in pain and a range of motion after arthroscopic surgery [113]. Furthermore, there was an improvement in low back pain after 5 min of intervention on the multifidus muscle in patients with acute low back pain [114]. In addition, the therapist and patient can see where the fascia is being stripped using ultrasound imaging, possibly because the patient may experience changes in muscle tension and pain, thus, increasing the level of expectations and satisfaction with the treatment. MFR in physical therapy (manual therapy) uses techniques systematized by Barnes and colleagues [115,116]. This is a technique that not only stretches the fascia but also unties and untwists it. Recently, self-myofascial release (SMFR) using foam rolling (FR) has become popular. The effects of FR include reducing delayed-onset muscle soreness after exercise [117], increasing the pain threshold, and making the pain less perceptible [118]. It has also been reported to improve the range of motion without decreasing exercise performance [119].

MFR has been applied to treat a wide range of disease areas, including tension-type headaches [120], postmenopausal venous insufficiency [121], and nonspecific chronic low back pain [122]. This is believed to be because MFR does not require joint movement and applies mild myofascial stretching and pressure, making it safe and applicable to many age groups and diseases.

Furthermore, Ichikawa et al. compared fascial gliding and flexibility of the vastus lateralis muscle with MFR and hot pack treatment and reported improved gliding and flexibility in MFR [123]. It was suggested that continuous lengthening and pressure were necessary to improve these. Furthermore, several reports have shown the effectiveness of MFR in combination with conventional therapies rather than as a standalone treatment [124,125]. For example, Ozsoy et al. reported that in elderly patients with nonspecific low back pain, the group that received MFR and trunk stabilization exercises showed a significant increase in MFR and trunk compared to the group that received only MFR. In addition, the results reported that trunk endurance and mobility improved in the group that received MFR and trunk stabilization exercises [126], suggesting that MFR in combination with conventional exercise therapy is effective.

Recently, scattered reports have shown the effectiveness of MFR in combination with conventional treatment for postoperative orthopedic patients [127,128]. Therefore, the Oswestry disability index (ODI) was examined at the beginning of the study, and a 1-month follow-up after the intervention was discontinued. The exercise was performed three times a week under the supervision of a physical therapist. The results showed that ODI improved significantly more with MFR than with SE alone, with a minimally clinically important difference of more than 10% [129,130]. The mechanism of these effects is suggested to be that MFR causes normalization of the apoptosis rate of damaged fascia, changes in cell morphology, and reorientation of fibroblasts [131], reducing fascial shortening and thickening, improving normal muscle length and flexibility of fascial tissue [102]. Furthermore, by activating the descending pain suppression system [132], it is speculated that pain modulation occurred and contributed to the improvement of ODI.

However, studies using MFR in the postoperative orthopedic setting are still limited, making generalized conclusions difficult to establish. Postoperative patients have been reported to experience tissue fibrosis during wound healing [133]. Kawanishi et al. reported improved pain [134] and walking ability [135] by improving subcutaneous tissue gliding in patients with femoral metaphyseal fractures. The effectiveness of a broad range of orthopedic postoperative therapies that focus on the fascia and the connective tissue as a whole needs to be clarified.

SMFR is used to prevent injuries and maintain performance in sports [136,137]. For example, in soccer, in the early period after half-time, both physical and cognitive performance is reduced, and the risk of injury increases [138]. Kaya et al. tested the effects of SMFR by replicating the running distance and half-time experienced during a game with soccer players of various levels [139]. When SMFR was performed during the recreated half-time period, the subsequent decrease in sprint performance was suppressed [139]. This effect, reported by Okamoto et al., was attributed to SMFR improving vascular endothelial function, increasing blood flow to the muscles targeted by SMFR, increasing the supply of oxygen and other vital nutrients, and promoting more efficient removal of metabolites [140]. Athletes, young and old, male and female, novice and elite, are prone to delayed-onset muscle soreness (DOMS) after intense regular exercise; DOMS results in fibrous tissue adhesions that limit joint range of motion [115,141,142]. Adhesions of fibrous tissue and fascia occur due to disease or injury and reduce joint ROM, muscle length, muscle endurance, and motor coordination [115,143,144]. Therefore, suppressing DOMS may also inhibit fascial adhesions. Static stretching (SS) is generally used before exercise to improve the range of motion and prevent injury. It is also believed to decrease force and power, making it difficult to use before exercise [145]. However, in a study in which SMFR was performed prior to exercise, it improved exercise performance without decreasing force and power, in addition to improving the range of motion [142,146,147]. SMFR has also been reported to reduce muscle fatigue before exercise [147,148]. Thus, the use of SMFR before exercise is thought to be effective in restoring range of motion, fatigue, and performance after exercise [146,147]. A systematic review investigating the effects of post-exercise massage reported that massage, such as MFR, stimulates the parasympathetic nervous system, indirectly enhances the immune system by improving local circulation, and decreases inflammatory cytokines [149]. SMFR, especially after exercise, has been shown to be beneficial for recovery after exercise-induced muscle damage (EIMD), DOMS, and other impairments of physical performance [150,151,152,153,154]. The improved performance after SMFR has also been reported to last up to 72 h [152,154]. SMFR has been reported to be a safe intervention used for performance (especially flexibility) and recovery from previous training and competition and can reduce DOMS [154,155]. These findings suggest that SMFR, as a routine practice before and after exercise, may help prevent fascial adhesions in athletes and reduce the incidence of injury.

When the fascia is restricted in any part, it causes stress and impairment in other areas [156], depending on the continuity of the myofascial structure, and reduces muscle flexibility [138]. This is a particularly important issue for athletes subjected to long-term repetitive strain, and using SMFR with FR is an important part of an athlete’s training. Therefore, SMFR with FR is recommended for inclusion in athletes’ training regarding performance loss, injury prevention, and recovery [138].

While some reports have shown the effectiveness of MFR, as mentioned earlier, there are several manual therapies for myofascial as follows: osteopathic soft tissue techniques [157], strain counter strain [158], myofascial trigger point therapy [159], muscle energy technique [160]. However, there is no evidence to suggest which manual therapy is optimal [161]. Furthermore, while MFR for chronic low back pain patients is effective for low back pain and ADL disability, the clinical significance of MFR is unclear [162]. Therefore, further studies are needed to clarify clinical relevance and build evidence in the future.

## 6. Conclusions

In myofascial tissue, normal fibroblasts are highly sensitive to physical stimuli. The transition from fibroblast to myofibroblast is influenced by mechanical stress and may result in gradual and irreversible tissue contraction as the contraction cycle is repeated many times. The acute inflammatory response of fascial tissue is usually brief and reversible but repeated identical postures, sports, and repetitive motion promote fibrosis via fibroblast overgrowth and collagen matrix deposition. Fibrosis around fascial tissue is translated into nerve mobility changes, resulting in dysfunction of central nociceptors. In addition, the thickening of the fascia increases the distance between surfaces, increasing HA viscosity. The increase in HA viscosity within the connective tissue inhibits the sliding of fascial collagen fibers between layers. Furthermore, MRI imaging allows the deep fascia to be observed and provides a broad and detailed assessment of fascial thickening and signal changes, as well as adjacent soft tissue and bone marrow edema; in contrast, ultrasound imaging allows dynamic observation of the fascia and is a useful tool for assessing the proximity and distal extent of the injury to determine treatment efficacy. The goal of myofascial release for myofascial dysfunction is to reduce pain and allow myofascial tissue mobility to function normally. Several manual therapies have been developed in addition to myofascial release; however, there is no evidence to suggest which manual therapy is best, and the effectiveness of a broad range of therapies targeting the entire connective tissue system needs to be clarified. Additionally, the effectiveness of a broad range of therapies that look at connective tissue as a whole also needs to be clarified.

## Data Availability

The data presented in this study are openly available in Bioengineering at Bioengineering|Special Issue: Physical Examination and Rehabilitation of Fasciae and Muscles in Sports Injuries (mdpi.com).

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
