# Peer review of "Response to Mechanical Properties and Physiological Challenges of Fascia: Diagnosis and Rehabilitative Therapeutic Intervention for Myofascial System Disorders"

_bioengineering, 2023, doi:10.3390/bioengineering10040474_

Round 1

Reviewer 1 Report

This review was comprehensive and well-organized. The reference sources are excellent. If you do not have these references I want you to consider including the following additional information regarding the mechanics of fascia and its relationship to the concept of tensegrity.

1. Fascia and living tensegrity considerations in: lower extremity and pelvic entrapment neuropathies

  • March 2021
  • International Journal of Anatomy and Research 9(1.2):7881-7885
  • DOI: 
  • 10.16965/ijar.2020.254

2. John Sharkey. Fascia and Tensegrity The Quintessence of a Unified Systems Conception. Int J Anat Appl Physiol. 2021;07(02):174-178.

3. Bordoni B, Myers T (February 24, 2020) A Review of the Theoretical Fascial Models: Biotensegrity, Fascintegrity, and Myofascial Chains. Cureus 12(2): e7092. DOI 10.7759/cureus.7092

    4. Dischiavi SL et al. (2018) Biotensegrity and myofascial chains: a global approach to an integrated kinetic chain. Medical Hypotheses 110, 90-96.
https://doi.org/10.1016/j.mehy.2017.11.008

Author Response

Comments and Suggestions for Authors

This review was comprehensive and well-organized. The reference sources are excellent. If you do not have these references I want you to consider including the following additional information regarding the mechanics of fascia and its relationship to the concept of tensegrity.

Thank you for your suggestion. We have accordingly changed the text in Section 2 and have included the references as instructed.

Line 72-82: The fact that the fascia can transmit tension far is the basis of the "biotensegrity" framework [31,32]. Biotensegrity is the application of the principle of tensegrity to the understanding of human movement, where tensegrity is an architectural principle according to which a structure (or tensegrity system) is stabilized by continuous tension with discontinuous compression and functions as a single structure [32]. As the tension in the fascia increases, the connective tissue can disperse the force around it and propagate it along the fascial system [31-34]. Forces passively imposed on the muscle by stretching are distributed throughout the tissue via the intramuscular connective tissue [33,34]. Fascia transmits tension, influences other muscles, plays a role in the proper coordination of body movements, and can reflect the direction of force vectors. 

  1. Fascia and living tensegrity considerations in: lower extremity and pelvic entrapment neuropathies. March 2021 International Journal of Anatomy and Research 2021, 9(1.2), 7881-7885, DOI: 10.16965/ijar.2020.254.
  2. Sharkey, J. Fascia and Tensegrity The Quintessence of a Unified Systems Conception. Int J Anat Appl Physiol. 2021, 07(02), 174-178.
  3. Bordoni, B; Myers, T. A Review of the Theoretical Fascial Models: Biotensegrity, Fascintegrity, and Myofascial Chains. Cureus. 2020, 12(2), e7092. DOI: 10.7759/cureus.7092.
  4. Dischiavi S.L.; Wright, A.A.; Hegedus, E.J.; Bleakley, C.M. (2018) Biotensegrity and myofascial chains: a global approach to an integrated kinetic chain. Medical Hypotheses. 2018, 110, 90-96. DOI:10.1016/j.mehy.2017.11.008

Reviewer 2 Report

The definition of fascial tissue, which was written in lines 56-60, seems to be incorrect, as the ECM is just extracellular molecular components, in general, we cannot be equal the fascial tissue to just the ECM. The whole contents in the section 2 ‘Molecular and Cytological Adaptations of Fascial Tissue’ need to be rewritten according to the correct definition of fascial tissue, including its main molecular components and main cell types. It would be better to show the more specific concept in the section title, as now the term ‘adaptations’ seems to be too general, and which is adapting to whom? Furthermore, such as, what molecular crosstalks? What bidirectional molecular interactions? What molecular chains? How plasticity in functional and structural? These are in a very general issues, and not be understanding if readers are not familiar, and actually they are sure to be not easily familiar with such topics.  

And more, the sections 3-7, these all may be factors influence to the physiological or pathological developments in fascial tissue, thus they could be umbrellaed under one more broad title, and if could, their content-describing should be closely linking into the above section 2, if the section 2 could be a functional and structural basis for understanding the fascial tissue.

Similarly, the sections 9-11 may also be put together under one title.

Author Response

Reviewer 2

Comments and Suggestions for Authors

The definition of fascial tissue, which was written in lines 56-60, seems to be incorrect, as the ECM is just extracellular molecular components, in general, we cannot be equal the fascial tissue to just the ECM. The whole contents in the section 2 ‘Molecular and Cytological Adaptations of Fascial Tissue’ need to be rewritten according to the correct definition of fascial tissue, including its main molecular components and main cell types. It would be better to show the more specific concept in the section title, as now the term ‘adaptations’ seems to be too general, and which is adapting to whom? Furthermore, such as, what molecular crosstalks? What bidirectional molecular interactions? What molecular chains? How plasticity in functional and structural? These are in a very general issues, and not be understanding if readers are not familiar, and actually they are sure to be not easily familiar with such topics.  

Thank you for this comment. Upon reflection, we agree with you and don't think we can equate fascial tissue with the ECM. Regarding molecular crosstalk and interactions, I have searched the cited literature and could not find any detailed data to explain this. I have, therefore, removed the text related to this. The opinion of reviewer 1 was also taken into consideration. The entire text of Section 2 has been changed accordingly.

Line 62-68:

  1. Role of fascial tissue and pathological reactions

Fascia is widely distributed from head to toe; it encases and permeates muscles, bones, blood vessels, nerves, and internal organs, constitutes various layers of different depths [25], and is a connective tissue composed of irregularly arranged collagen fibers, clearly different from the regularly arranged collagen fibers found in tendons, ligaments, or periosteal sheets [26]. Additionally, it supports important functions of the human body, such as posture, movement, and homeostasis [25-28], and also contains various sensory receptors for proprioception, nociception, and even hormones [28].

Willard F.H.; Vleeming A.; Schuenke M.D.; Danneels L.; Schleip R. The thoracolumbar fascia: anatomy, function and clinical considerations. J Anat. 2012, 221(6), 507–36. doi: 10.1111/j.1469-7580.2012.01511.x.

Langevin, H.M.; Keely, P.; Mao, J.; Hodge, L.M.; Schleip, R.; Deng, G.; Hinz, B.; Swartz, M.A.; De Valois, B.A.; Zick, S.; Findley, T. Connecting (T)issues: How Research in Fascia Biology Can Impact Integrative Oncology. Cancer Res. 2016, 76(21), 6159–6162. doi: 10.1158/0008-5472.CAN-16-0753.

Barker, P.J.; Briggs, C.A. Attachments of the posterior layer of lumbar fascia. Spine (Phila Pa 1976). 1999, 24(17), 1757–64. doi: 10.1097/00007632-199909010-00002.

Nordez, A.; Gross, R.; Andrade, R.; Le Sant, G.; Freitas, S.; Ellis, R.; McNair, P.J.; Hug, F. Non-Muscular Structures Can Limit the Maximal Joint Range of Motion during Stretching. Sports Med. 2017, 47(10), 1925–1929. doi: 10.1007/s40279-017-0703-5.

Line 72-82: The fact that the fascia can transmit tension far is the basis of the "biotensegrity" framework [31,32]. Biotensegrity is the application of the principle of tensegrity to the understanding of human movement, where tensegrity is an architectural principle according to which a structure (or tensegrity system) is stabilized by continuous tension with discontinuous compression and functions as a single structure [32]. As the tension in the fascia increases, the connective tissue can disperse the force around it and propagate it along the fascial system [31-34]. Forces passively imposed on the muscle by stretching are distributed throughout the tissue via the intramuscular connective tissue [33,34]. Fascia transmits tension, influences other muscles, plays a role in the proper coordination of body movements, and can reflect the direction of force vectors. 

  1. Fascia and living tensegrity considerations in: lower extremity and pelvic entrapment neuropathies. March 2021 International Journal of Anatomy and Research 2021, 9(1.2), 7881-7885, DOI: 10.16965/ijar.2020.254.
  2. Sharkey, J. Fascia and Tensegrity The Quintessence of a Unified Systems Conception. Int J Anat Appl Physiol. 2021, 07(02), 174-178.
  3. Bordoni, B; Myers, T. A Review of the Theoretical Fascial Models: Biotensegrity, Fascintegrity, and Myofascial Chains. Cureus. 2020, 12(2), e7092. DOI: 10.7759/cureus.7092.
  4. Dischiavi S.L.; Wright, A.A.; Hegedus, E.J.; Bleakley, C.M. (2018) Biotensegrity and myofascial chains: a global approach to an integrated kinetic chain. Medical Hypotheses. 2018, 110, 90-96. DOI:10.1016/j.mehy.2017.11.008

And more, the sections 3-7, these all may be factors influence to the physiological or pathological developments in fascial tissue, thus they could be umbrellaed under one more broad title, and if could, their content-describing should be closely linking into the above section 2, if the section 2 could be a functional and structural basis for understanding the fascial tissue.

Thank you for pointing this out. Paragraphs 3-7 encompassed factors affecting the pathological development of the fascial tissue.

Line 101: 3. Factors influencing the pathological development of fascia tissue (nerve, disorder, aging, sex hormone)

Similarly, the sections 9-11 may also be put together under one title.

Line:276:5. Myofascial release (MFR) for muscle and fascia dysfunction

Reviewer 3 Report

This paper reviews the molecular level responses of the mechanical properties of fascia and the related studies on other physiological responses. The results suggest that fascial injury results in a significant decrease in performance during exercise and may lead to musculoskeletal disease and the development of persistent underlying pain. The fascia is significantly different from the regular array of collagen fibers found in tendons, ligaments, or periosteum, and mechanical changes in the fascia can cause changes in its connective tissue, which can cause pain. Although these mechanical changes cause inflammation associated with mechanical load, they are also affected by biochemical influences such as aging, sex hormones and obesity. In response, I have the following questions:

1.The author points out that in the process of physical examination, the fascial injury is often not taken into account, so how to check the fascial injury in modern medicine.

2.The author's article shows that the fascia damage can significantly reduce the performance of sports, so for different degrees of damage, how can we effectively repair.

3.The author's article shows that excessive exercise can lead to muscle fibrosis, but it is difficult to find the critical point between exercise and rest. Whether we can find this critical point through the perception of muscle soreness, suggest that the author look up this part of the data.

4.As mentioned in the author's article, substance P can repair and increase collagen in fascia. What methods can we use to promote the production of substance P? It is suggested that the author look for relevant information.

5.The author mentioned in the author's article that inflammatory cytokines entering the blood will damage the fascia, so what ways can we reduce inflammatory cytokines in the blood at ordinary times? It is suggested that the author add this part.

Author Response

Reviewer 3

Comments and Suggestions for Authors

This paper reviews the molecular level responses of the mechanical properties of fascia and the related studies on other physiological responses. The results suggest that fascial injury results in a significant decrease in performance during exercise and may lead to musculoskeletal disease and the development of persistent underlying pain. The fascia is significantly different from the regular array of collagen fibers found in tendons, ligaments, or periosteum, and mechanical changes in the fascia can cause changes in its connective tissue, which can cause pain. Although these mechanical changes cause inflammation associated with mechanical load, they are also affected by biochemical influences such as aging, sex hormones and obesity. In response, I have the following questions:

Thank you for your comments. We have responded to all the remarks and have made changes accordingly.

1.The author points out that in the process of physical examination, the fascial injury is often not taken into account, so how to check the fascial injury in modern medicine.

Thank you for pointing this out. We have accordingly made changes to the text based on the references cited.

Line 41-48: For example, acute pain associated with the groin is common in athletes injured by overuse. Acute strains have been shown to occur at the musculotendinous junction, particularly in the adductor longus, rectus femoris, and iliopsoas muscles [16]. X-rays and MRIs are performed to rule out the diagnosis of serious diseases, such as fractures [17-20], and if serious diseases are ruled out, pain provocation tests are performed using palpation, stretching, and resistance testing [21,22]. However, these evaluation methods focus on the joint range of motion and muscle strength and do not consider the assessment of disorders of the fascia [23,24].

  1. Serner, A., Weir, A., Tol, J.L., Thorborg, K., Roemer, F., Guermazi, A., Yamashiro, E.; Hölmich, P. Characteristics of acute groin injuries in the adductor muscles: a detailed MRI study in athletes. Scand J Med Sci Sports. 2018, 28, 667-676. DOI: 10.1111/sms.12936.
  2. Davies, A.G.; Clarke, A.W.; Gilmore, J.; Wotherspoon, M.; Connell, D.A. Imaging of groin pain in the athlete. Skeletal Radiol. 2010, 39, 629-644. DOI:10.1007/s00256-009-0768-9.
  3. Knapik, J.J.; Reynolds, K.L.; Hoedebecke, K.L. Stress fractures: etiology, epidemiology, diagnosis, treatment, and prevention. J Spec Oper Med. 2017, 17, 120-130.
  4. Armfield, D.R.; Towers, J.D.; Robertson, D.D. Radiographic and MR imaging of the athletic hip. Clin Sports Med. 2006, 25, 211-239. DOI:10.1016/j.csm.2005.12.009.
  5. Georgiadis, A.G.; Zaltz, I. Slipped capital femoral epiphysis: how to evaluate with a review and update of treatment. Pediatr Clin North Am. 2014, 61, 1119-1135. DOI: 10.1016/j. pcl.2014.08.001.
  6. Serner, A.; Tol, J.L.; Jomaah, N.; Weir, A.; Whiteley, R.; Thorborg, K.; Robinson, M.; Hölmich, P. Diagnosis of acute groin injuries: a prospective study of 110 athletes. Am J Sports Med. 2015, 43, 1857-1864. DOI: 10.1177/0363546515585123.
  7. Serner, A.; Weir, A.; Tol, J.L.; Thorborg, K.; Roemer, F.; Guermazi, A.; Hölmich, P. Can standardised clinical examination of athletes with acute groin injuries predict the presence and location of MRI findings? Br J Sports Med. 2016, 50, 1541-1547. DOI: 10.1136/bjsports-2016-096290.
  8. Thorborg, K.; Branci, S.; Nielsen, M.P.; Tang, L.; Nielsen, M.B.; Hölmich, P. Eccentric and isometric hip adduction strength in male soccer players with and without adductor-related groin pain: an assessor-blinded comparison. Orthop J Sports Med. 2014, 2, 2325967114521778. DOI:10.1177/2325967114521778.
  9. Reiman, M.P.; Thorborg, K. Clinical examination and physical assessment of hip joint-related pain in athletes. Int J Sports Phys Ther. 2014, 9, 737-755.

2.The author’s article shows that the fascia damage can significantly reduce the performance of sports, so for different degrees of damage, how can we effectively repair.

Thank you for pointing this out. We have accordingly made changes to the text based on the references cited.

Line 363-385: Athletes, young and old, male and female, novice and elite, are prone to delayed onset muscle soreness (DOMS) after intense regular exercise; DOMS results in fibrous tissue adhesions that limit joint range of motion [144-146]. Adhesions of fibrous tissue and fascia occur due to disease or injury and reduce joint ROM, muscle length, muscle endurance, and motor coordination [144,147,148]. Therefore, suppressing DOMS may also inhibit fascial adhesions. Static stretching (SS) is generally used before exercise to improve the range of motion and prevent injury. It is also believed to decrease force and power, making it difficult to use before exercise [149]. However, in a study in which SMFR was performed prior to exercise, it improved exercise performance without decreasing force and power, in addition to improving the range of motion [146,150,151]. SMFR has also been reported to reduce muscle fatigue before exercise [152,153]. Thus, the use of SMFR before exercise is thought to be effective in restoring range of motion, fatigue, and performance after exercise [151,152]. A systematic review investigating the effects of post-exercise massage reported that massage, such as MFR, stimulates the parasympathetic nervous system, indirectly enhances the immune system by improving local circulation, and decreases inflammatory cytokines [154]. SMFR, especially after exercise, has been shown to be beneficial for recovery after exercise-induced muscle damage (EIMD), DOMS, and other impairments of physical performance [155-157]. The improved performance after SMFR has also been reported to last up to 72 hours [156,158]. SMFR has been reported to be a safe intervention used for performance (especially flexibility) and recovery from previous training and competition and can reduce DOMS [159,160]. These findings suggest that SMFR as a routine practice before and after exercise may help prevent fascial adhesions in athletes and reduce the incidence of injury.

  1. Barnes, M.F. The basic science of myofascial release: morphologic change in connective tissue. J. Bodyw. Mov. Ther. 1997; 1, 231Y8.
  2. Cheung, K.; Hume, P.; Maxwell, L. Delayed onset muscle soreness : treatment strategies and performance factors. Sports Med. 2003, 33(2), 145-64. DOI:10.2165/00007256-200333020-00005. PMID: 12617692.
  3. Halperin, I.; Aboodarda, S.J.; Button, D.C.; Andersen, L.L.; Behm, D.G. Roller massager improves range of motion of plantar flexor muscles without subsequent decreases in force parameters. Int J Sports Phys Ther. 2014, 9(1), 92-102. PMID: 24567860; PMCID: PMC3924613.
  4. Curran, P.F.; Fiore, R.D.; Crisco, J.J. A comparison of the pressure exerted on soft tissue by 2 myofascial rollers. J Sport Rehabil. 2008, 17(4), 432-42. DOI:10.1123/jsr.17.4.432. PMID: 19160916.
  5. Swann, E.; Graner, S.J. Uses of manual-therapy techniques in pain management. Athl. Ther. Today. 2002, 7, 14Y7.
  6. Behm, D.G.; Chaouachi, A. A review of the acute effects of static and dynamic stretching on performance. Eur J Appl Physiol. 2011, 111(11), 2633-51. DOI:10.1007/s00421-011-1879-2. PMID: 21373870.
  7. MacDonald, G.Z.; Penney, M.D.; Mullaley, M.E.; Cuconato, A.L.; Drake, C.D.; Behm, D.G.; Button, D.C. An acute bout of self-myofascial release increases range of motion without a subsequent decrease in muscle activation or force. J Strength Cond Res. 2013, 27(3), 812-21. DOI:10.1519/JSC.0b013e31825c2bc1. PMID: 22580977.
  8. Macdonald, G.Z.; Button, D.C.; Drinkwater, E.J.; Behm, D.G. Foam rolling as a recovery tool after an intense bout of physical activity. Med Sci Sports Exerc. 2014, 46(1), 131-42. DOI:10.1249/MSS.0b013e3182a123db. PMID: 24343353.
  9. Healey, K.C.; Hatfield, D.L.; Blanpied, P.; Dorfman, L.R.; Riebe, D. The effects of myofascial release with foam rolling on performance. J Strength Cond Res. 2014, 28(1), 61-8. DOI:10.1519/JSC.0b013e3182956569. PMID: 23588488.
  10. Schroeder, A.N.; Best, T.M. Is self myofascial release an effective preexercise and recovery strategy? A literature review. Curr Sports Med Rep. 2015;14(3):200-8. DOI: 10.1249/JSR.0000000000000148. Erratum in: Curr Sports Med Rep. 2015, 14(5), 352. PMID: 25968853.
  11. Tejero-Fernández, V.; Membrilla-Mesa, M.; Galiano-Castillo, N.; Arroyo-Morales, M. Immunological effects of massage after exercise: A systematic review. Phys Ther Sport. 2015, 16(2), 187-92. DOI:10.1016/j.ptsp.2014.07.001. PMID: 25116861.
  12. Hendricks, S.; Hill, H.; Hollander, S.D.; Lombard, W.; Parker, R. Effects of foam rolling on performance and recovery: A systematic review of the literature to guide practitioners on the use of foam rolling. J Bodyw Mov Ther. 2020, 24(2), 151-174. DOI:10.1016/j.jbmt.2019.10.019. PMID: 32507141.
  13. Skinner, B.; Moss, R.; Hammond, L. A systematic review and meta-analysis of the effects of foam rolling on range of motion, recovery and markers of athletic performance. J Bodyw Mov Ther. 2020, 24(3), 105-122. DOI:10.1016/j.jbmt.2020.01.007. PMID: 32825976.
  14. Wiewelhove, T.; Döweling, A.; Schneider, C.; Hottenrott, L.; Meyer, T.; Kellmann, M.; Pfeiffer, M.; Ferrauti, A. A Meta-Analysis of the Effects of Foam Rolling on Performance and Recovery. Front Physiol. 2019, 10, 376. DOI:10.3389/fphys.2019.00376. PMID: 31024339; PMCID: PMC6465761.
  15. Hughes, G.A.; Ramer, L.M. duration of myofascial rolling for optimal recovery, range of motion, and performance: a systematic review of the literature. Int J Sports Phys Ther. 2019, 14(6), 845-859. PMID: 31803517; PMCID: PMC6878859.
  16. Ferreira, R.M.; Martins, P.N.; Goncalves, R.S. Effects of Self-myofascial Release Instruments on Performance and Recovery: An Umbrella Review. Int J Exerc Sci. 2022, 15(3), 861-883. PMID: 35991349; PMCID: PMC9362891.
  17. Jay, K.; Sundstrup, E.; Søndergaard, S.D.; Behm, D.; Brandt, M.; Særvoll, C.A.; Jakobsen, M.D.; Andersen, L.L. Specific and cross over effects of massage for muscle soreness: randomized controlled trial. Int J Sports Phys Ther. 2014, 9(1), 82-91. PMID: 24567859; PMCID: PMC3924612.

3.The author's article shows that excessive exercise can lead to muscle fibrosis, but it is difficult to find the critical point between exercise and rest. Whether we can find this critical point through the perception of muscle soreness, suggest that the author look up this part of the data.

Thank you for pointing this out. We have accordingly made changes to the text based on the references cited.

Line 288-296: Training is necessary to improve performance, especially in athletes; however, inadequate rest periods can cause high-frequency, high-intensity training that leads to continued pointless training [111]. Maladaptive training before tissue recovery and rebuilding can lead to the accumulation of microdamage in affected tissues, resulting in overuse injuries, and thereby, compromising the athletes’ competitive performance due to pain and dysfunction [112]. As evidenced by the fact that 39% of athletes experience unexplained musculoskeletal pain weekly [113], athletes may continue to train despite the risk of disability. Since it is difficult to determine training at the appropriate load and rest periods, we believe that MFR, as discussed below, can reduce the risk of overuse injury.

  1. Soligard, T.; Schwellnus, M.; Alonso, J.M.; Bahr, R.; Clarsen, B.; Dijkstra, H.P.; Gabbett, T.; Gleeson, M.; Hägglund, M.; Hutchinson, M.R.; Janse van Rensburg, C.; Khan, K.M.; Meeusen, R.; Orchard, J.W.; Pluim, B.M.; Raftery, M.; Budgett, R.; Engebretsen, L. How much is too much? (Part 1) International Olympic Committee consensus statement on load in sport and risk of injury. Br J Sports Med. 2016, 50(17), 1030-41. DOI:10.1136/bjsports-2016-096581. PMID: 27535989.
  2. Bahr, R. No injuries, but plenty of pain? On the methodology for recording overuse symptoms in sports. Br J Sports Med. 2009, 43(13), 966-72. DOI:10.1136/bjsm.2009.066936. PMID: 19945978.
  3. Clarsen, B.; Myklebust, G.; Bahr, R. Development and validation of a new method for the registration of overuse injuries in sports injury epidemiology: the Oslo Sports Trauma Research Centre (OSTRC) overuse injury questionnaire. Br J Sports Med. 2013, 47(8), 495-502. DOI:10.1136/bjsports-2012-091524. PMID: 23038786.

4.As mentioned in the author’s article, substance P can repair and increase collagen in fascia. What methods can we use to promote the production of substance P? It is suggested that the author look for relevant information.

Thank you for pointing this out. We have accordingly made changes to the text based on the references cited.

Line 140-145: Prolonged or repeated loading results in persistent inflammation and the prolonged presence of macrophages and cytotoxic cytokines in and around the tissue [64,65]. Eventually, tissue damage progresses, and overproduction of cytokines is triggered. This overproduction of cytokines maintains sensitization of nociceptive afferents and increases the production and release of substance P [66].

  1. Gao, H.G.; Fisher, P.W.; Lambi, A.G.; Wade, C.K.; Barr-Gillespie, A.E.; Popoff, S.N.; Barbe, M.F. Increased serum and musculotendinous fibrogenic proteins following persistent low-grade inflammation in a rat model of long-term upper extremity overuse. PLOS ONE. 2013, 8, e71875. DOI:10.1371/journal.pone.0071875.
  2. Barbe, M.F.; Gallagher, S.; Popoff, S.N. Serum biomarkers as predictors of stage of work-related musculoskeletal disorders. J Am Acad Orthop Surg. 2013, 21, 644–646. DOI:10.5435/JAAOS-21-10-644.
  3. Frara, N.; Fisher, P.W.; Zhao, Y.; Tarr, J.T.; Amin, M.; Popoff, S.N.; Barbe, M.F. Substance P increases CCN2 dependent on TGF-beta yet Collagen Type I via TGF-beta1 dependent and independent pathways in tenocytes. Connect Tissue Res. 2018, 59, 30–44. DOI:10.1080/03008207.2017.1297809.

Line 155-159: The following methods have been reported to decrease inflammatory cytokines. Early treatment with anti-inflammatory agents can prevent or reduce pain induced by TNF signaling and decrease downstream collagen production [69]. Stretching of the fascial tissue can promote the resolution of inflammation both in vivo and in vitro, and manual therapy can prevent overuse-induced fibrosis in some fascial tissues [70,71].

  1. Kasper, D.L.; Fauci, A.S.; Hauser, S.L.; Longo, D.. L. 1. Jameson J. L., & Loscalzo J. Harrison’s Principles of Internal Medicine, 20e; New York; 2018. pp. 222–223, 2637–2639, 2644–2645.
  2. Abdelmagid, S.M.; Barr, A.E.; Rico, M.; Amin, M.; Litvin, J.; Popoff, S.N.; Safadi, F.F.; Barbe, M.F. Performance of repetitive tasks induces decreased grip strength and increased fibrogenic proteins in skeletal muscle: role of force and inflammation. PLoS One2012, 7, e38359 DOI:10.1371/journal.pone.0038359.
  3. Berrueta, L.; Muskaj, I.; Olenich, S.; Butler, T.; Badger, G.J.; Colas, R.A.; Spite, M.; Serhan, C.N.; Langevin, H.M. Stretching impacts inflammation resolution in connective tissue. J Cell Physiol2016, 231, 1621–7. DOI:10.1002/jcp.25263.

Round 2

Reviewer 2 Report

This revised manuscript makes a large improvement for the content arrangement, thus read more compact and logically. 

Reviewer 3 Report

Overall achieving readability, details can be further optimized.